# Improved Iteration Complexity Bounds of Cyclic Block Coordinate Descent for Convex Problems

**Ruoyu Sun,**[*] **Mingyi Hong**[†‡]

## Abstract

The iteration complexity of the block-coordinate descent (BCD) type algorithm has been under extensive investigation. It was recently shown that for convex problems the classical cyclic BCGD (block coordinate gradient descent) achieves an $\mathcal{O}(1/r)$ complexity ($r$ is the number of passes of all blocks). However, such bounds are at least linearly depend on $K$ (the number of variable blocks), and are at least $K$ times worse than those of the gradient descent (GD) and proximal gradient (PG) methods. In this paper, we close such theoretical performance gap between cyclic BCD and GD/PG. First we show that for a family of quadratic nonsmooth problems, the complexity bounds for cyclic Block Coordinate Proximal Gradient (BCPG), a popular variant of BCD, can match those of the GD/PG in terms of dependency on $K$ (up to a $\log^2(K)$ factor). Second, we establish an improved complexity bound for Coordinate Gradient Descent (CGD) for general convex problems which can match that of GD in certain scenarios. Our bounds are sharper than the known bounds as they are always at least $K$ times worse than GD. Our analyses do not depend on the update order of block variables inside each cycle, thus our results also apply to BCD methods with random permutation (random sampling without replacement, another popular variant).

## 1 Introduction

Consider the following convex optimization problem

$$\min \quad f(x) = g(x_1, \cdots, x_K) + \sum_{k=1}^{K} h_k(x_k), \quad \text{s.t.} \quad x_k \in X_k, \ \forall \, k = 1, \cdots K, \qquad (1)$$

where $g : X \to \mathbb{R}$ is a convex smooth function; $h : X \to \mathbb{R}$ is a convex lower semi-continuous possibly nonsmooth function; $x_k \in X_k \subseteq \mathbb{R}^N$ is a block variable. A very popular method for solving this problem is the so-called block coordinate descent (BCD) method [5], where each time a single block variable is optimized while the rest of the variables remain fixed. Using the classical cyclic block selection rule, the BCD method can be described below.

---

**Algorithm 1: The Cyclic Block Coordinate Descent (BCD)**

At each iteration $r + 1$, update the variable blocks by:

$$x_k^{(r)} \in \min_{x_k \in X_k} \ g\left(x_k, w_{-k}^{(r)}\right) + h_k(x_k), \ k = 1, \cdots, K. \qquad (2)$$

---

---

[*]Department of Management Science and Engineering, Stanford University, Stanford, CA. `ruoyu@stanford.edu`

[†]Department of Industrial & Manufacturing Systems Engineering and Department of Electrical & Computer Engineering, Iowa State University, Ames, IA, `mingyi@iastate.edu`

[‡]The authors contribute equally to this work.

where we have used the following short-handed notations:

$$w_k^{(r)} := \left[ x_1^{(r)}, \cdots, x_{k-1}^{(r)}, x_k^{(r-1)}, x_{k+1}^{(r-1)}, \cdots, x_K^{(r-1)} \right], \ k = 1, \cdots, K,$$

$$w_{-k}^{(r)} := \left[ x_1^{(r)}, \cdots, x_{k-1}^{(r)}, x_{k+1}^{(r-1)}, \cdots, x_K^{(r-1)} \right], \ k = 1, \cdots, K,$$

$$x_{-k} := [x_1, \cdots, x_{k-1}, x_{k+1}, \cdots, x_K].$$

The convergence analysis of the BCD has been extensively studied in the literature, see [5, 14, 19, 15, 4, 7, 6, 10, 20]. For example it is known that for smooth problems (i.e. $f$ is continuous differentiable but possibly nonconvex, $h = 0$), if each subproblem has a unique solution and $g$ is non-decreasing in the interval between the current iterate and the minimizer of the subproblem (one special case is per-block strict convexity), then every limit point of $\{x^{(r)}\}$ is a stationary point [5, Proposition 2.7.1]. The authors of [6, 19] have derived relaxed conditions on the convergence of BCD. In particular, when problem (1) is convex and the level sets are compact, the convergence of the BCD is guaranteed without requiring the subproblems to have unique solutions [6]. Recently Razaviyayn *et al* [15] have shown that the BCD converges if each subproblem (2) is solved inexactly, by way of optimizing certain surrogate functions.

Luo and Tseng in [10] have shown that when problem (1) satisfies certain additional assumptions such as having a smooth composite objective and a polyhedral feasible set, then BCD converges linearly without requiring the objective to be strongly convex. There are many recent works on showing iteration complexity for randomized BCGD (block coordinate gradient descent), see [17, 12, 8, 16, 9] and the references therein. However the results on the classical cyclic BCD is rather scant. Saha and Tewari [18] show that the cyclic BCD achieves sublinear convergence for a family of special LASSO problems. Nutini *et al* [13] show that when the problem is strongly convex, unconstrained and smooth, BCGD with certain Gauss-Southwell block selection rule could be faster than the randomized rule. Recently Beck and Tetruashvili show that cyclic BCGD converges sublinearly if the objective is smooth. Subsequently Hong *et al* in [7] show that such sublinear rate not only can be extended to problems with nonsmooth objective, but is true for a large family of BCD-type algorithm (with or without per-block exact minimization, which includes BCGD as a special case). When each block is minimized exactly and when there is no per-block strong convexity, Beck [2] proves the sublinear convergence for certain 2-block convex problem (with only one block having Lipschitzian gradient). It is worth mentioning that all the above results on cyclic BCD can be used to prove the complexity for a popular *randomly permuted* BCD in which the blocks are randomly sampled without replacement.

To illustrate the rates developed for the cyclic BCD algorithm, let us define $X^*$ to be the optimal solution set for problem (1), and define the constant

$$R_0 := \max_{x \in X} \max_{x^* \in X^*} \left\{ \|x - x^*\| \mid f(x) \leq f(x^{(0)}) \right\}. \tag{3}$$

Let us assume that $h_k(x_k) \equiv 0$, $X_k = \mathbb{R}^N$, $\forall\, k$ for now, and assume that $g(\cdot)$ has Lipschitz continuous gradient:

$$\|\nabla g(x) - \nabla g(z)\| \leq L\|x - z\|, \ \forall\, x, z \in X. \tag{4}$$

Also assume that $g(\cdot, x_{-k})$ has Lipschitz continuous gradient with respect to each $x_k$, i.e.,

$$\|\nabla_k g(x_k, x_{-k}) - \nabla_k g(v_k, x_{-k})\| \leq L_k\|x_k - v_k\|, \ \forall\, x, v \in X, \ \forall\, k. \tag{5}$$

Let $L_{\max} := \max_k L_k$ and $L_{\min} := \min_k L_k$. It is known that the cyclic BCPG has the following iteration complexity [4, 7] [1]

$$\Delta_{\text{BCD}}^{(r)} := f(x^{(r)}) - f^* \leq CL_{\max}(1 + KL^2/L_{\min}^2)R_0^2\frac{1}{r}, \quad \forall\, r \geq 1, \tag{6}$$

where $C > 0$ is some constant independent of problem dimension. Similar bounds are provided for cyclic BCD in [7, Theorem 6.1]. In contrast, it is well known that when applying the classical

gradient descent (GD) method to problem (1) with the constant stepsize $1/L$, we have the following rate estimate [11, Corollary 2.1.2]

$$\Delta_{\text{GD}}^{(r)} := f(x^{(r)}) - f(x^*) \le \frac{2\|x^{(0)} - x^*\|^2 L}{r+4} \le \frac{2R_0^2 L}{r+4}, \quad \forall\, r \ge 1, \ \forall\, x^* \in X^*. \tag{7}$$

Note that unlike (6), here the constant in front of the $1/(r+4)$ term is *independent* of the problem dimension. In fact, the ratio of the bound given in (6) and (7) is

$$\frac{CL_{\max}}{L}(1 + KL^2/L_{\min}^2)\frac{r+4}{r}$$

which is at least in the order of $K$. For big data related problems with over millions of variables, a multiplicative constant in the order of $K$ can be a serious issue. In a recent work by Saha and Tewari [18], the authors show that for a LASSO problem with special data matrix, the rate of cyclic BCD (with special initialization) is indeed $K$-independent. Unfortunately, such a result has not yet been extended to any other convex problems. An open question posed by a few authors [4, 3, 18] are: is such a $K$ factor gap intrinsic to the cyclic BCD or merely an artifact of the existing analysis?

## 2 Improved Bounds of Cyclic BCPG for Nonsmooth Quadratic Problem

In this section, we consider the following nonsmooth quadratic problem

$$\min\ f(x) := \frac{1}{2}\left\|\sum_{k=1}^{K} A_k x_k - b\right\|^2 + \sum_{k=1}^{K} h_k(x_k), \quad \text{s.t. } x_k \in X_k, \ \forall\, k \tag{8}$$

where $A_k \in \mathbb{R}^{M \times N}$; $b \in \mathbb{R}^M$; $x_k \in \mathbb{R}^N$ is the $k$th block coordinate; $h_k(\cdot)$ is the same as in (1). Note the blocks are assumed to have equal dimension for simplicity of presentation. Define $A := [A_1, \cdots, A_k] \in \mathbb{R}^{M \times KN}$. For simplicity, we have assumed that all the blocks have the same size. Problem (8) includes for example LASSO and group LASSO as special cases.

We consider the following cyclic BCPG algorithm.

---
**Algorithm 2: The Cyclic Block Coordinate Proximal Gradient (BCPG)**

At each iteration $r+1$, update the variable blocks by:

$$x_k^{(r+1)} = \arg\min_{x_k \in X_k}\ g(w_k^{(r+1)}) + \left\langle \nabla_k g\left(w_k^{(r+1)}\right), x_k - x_k^{(r)}\right\rangle + \frac{P_k}{2}\left\|x_k - x_k^{(r)}\right\|^2 + h_k(x_k)$$

$$\tag{9}$$
---

Here $P_k$ is the inverse of the stepsize for $x_k$, which satisfies

$$P_k \ge \lambda_{\max}\left(A_k^T A_k\right) = L_k, \ \forall\, k. \tag{10}$$

Define $P_{\max} := \max_k P_k$ and $P_{\min} = \min_k P_k$. Note that for the least square problem (smooth quadratic minimization, i.e. $h_k \equiv 0, \forall\, k$), BCPG reduces to the widely used BCGD method.

The optimality condition for the $k$th subproblem is given by

$$\left\langle \nabla_k g(w_k^{(r+1)}) + P_k(x_k^{(r+1)} - x_k^{(r)}), x_k - x_k^{(r+1)}\right\rangle + h_k(x_k) - h_k(x_k^{(r+1)}) \ge 0, \ \forall\, x_k \in X_k. \tag{11}$$

In what follows we show that the cyclic BCPG for problem (8) achieves a complexity bound that only dependents on $\log^2(NK)$, and apart from such log factor it is at least $K$ times better than those known in the literature. Our analysis consists of the following three main steps:

1. Estimate the descent of the objective after each BCPG iteration;
2. Estimate the cost yet to be minimized (cost-to-go) after each BCPG iteration;
3. Combine the above two estimates to obtain the final bound.

First we show that the BCPG achieves the sufficient descent.

**Lemma 2.1.** *We have the following estimate of the descent when using the BCPG:*

$$f(x^{(r)}) - f(x^{(r+1)}) \geq \sum_{k=1}^{K} \frac{P_k}{2} \|x_k^{(r+1)} - x_k^{(r)}\|^2. \tag{12}$$

**Proof.** We have the following series of inequalities

$$f(x^{(r)}) - f(x^{(r+1)})$$

$$= \sum_{k=1}^{K} f(w_k^{(r+1)}) - f(w_{k+1}^{(r+1)})$$

$$\geq \sum_{k=1}^{K} f(w_k^{(r+1)}) - \left( g(w_k^{(r+1)}) + h_k(x_k^{(r+1)}) + \left\langle \nabla_k g(w_k^{(r+1)}), x_k^{(r+1)} - x_k^{(r)} \right\rangle + \frac{P_k}{2} \left\| x_k^{(r+1)} - x_k^{(r)} \right\|^2 \right)$$

$$= \sum_{k=1}^{K} h_k(x_k^{(r)}) - h_k(x_k^{(r+1)}) - \left( \left\langle \nabla_k g \left( w_k^{(r+1)} \right), x_k^{(r+1)} - x_k^{(r)} \right\rangle + \frac{P_k}{2} \left\| x_k^{(r+1)} - x_k^{(r)} \right\|^2 \right)$$

$$\geq \sum_{k=1}^{K} \frac{P_k}{2} \|x_k^{(r+1)} - x_k^{(r)}\|^2.$$

where the second inequality uses the optimality condition (11). **Q.E.D.**

To proceed, let us introduce two matrices $\widetilde{P}$ and $\widetilde{A}$ given below, which have dimension $K \times K$ and $MK \times NK$, respectively

$$\widetilde{P} := \begin{bmatrix} P_1 & 0 & 0 & \cdots & 0 & 0 \\ 0 & P_2 & 0 & \cdots & 0 & 0 \\ \vdots & \vdots & \vdots & \cdots & \vdots & \vdots \\ 0 & 0 & 0 & \cdots & 0 & P_K \end{bmatrix}, \quad \widetilde{A} := \begin{bmatrix} A_1 & 0 & 0 & \cdots & 0 & 0 \\ 0 & A_2 & 0 & \cdots & 0 & 0 \\ \vdots & \vdots & \vdots & \cdots & \vdots & \vdots \\ 0 & 0 & 0 & \cdots & 0 & A_K \end{bmatrix}.$$

By utilizing the definition of $P_k$ in (10) we have the following inequalities (the second inequality comes from [12, Lemma 1])

$$\widetilde{P} \otimes I_N \succeq \widetilde{A}^T \widetilde{A}, \quad K \widetilde{A}^T \widetilde{A} \succeq A^T A \tag{13}$$

where $I_N$ is the $N \times N$ identity matrix and the notation "$\otimes$" denotes the Kronecker product.

Next let us estimate the cost-to-go.

**Lemma 2.2.** *We have the following estimate of the optimality gap when using the BCPG:*

$$\Delta^{(r+1)} := f(x^{(r+1)}) - f(x^*)$$

$$\leq R_0 \log(2NK) \left( L/\sqrt{P_{\min}} + \sqrt{P_{\max}} \right) \left\| (x^{(r+1)} - x^{(r)})(\widetilde{P}^{1/2} \otimes I_N) \right\| \tag{14}$$

Our third step combines the previous two steps and characterizes the iteration complexity. This is the main result of this section.

**Theorem 2.1.** *The iteration complexity of using BCPG to solve (8) is given below.*

1. *When the stepsizes are chosen conservatively as $P_k = L, \forall k$, we have*

$$\Delta^{(r+1)} \leq 3 \max \left\{ \Delta^0, 4 \log^2(2NK)L \right\} \frac{R_0^2}{r+1} \tag{15}$$

2. *When the stepsizes are chosen as $P_k = \lambda_{\max}(A_k^T A_k) = L_k, \forall k$. Then we have*

$$\Delta^{(r+1)} \leq 3 \max \left\{ \Delta^0, 2 \log^2(2NK) \left( L_{\max} + \frac{L^2}{L_{\min}} \right) \right\} \frac{R_0^2}{r+1} \tag{16}$$

*In particular, if the problem is smooth and unconstrained, i.e., when $h \equiv 0$, and $X_k = \mathbb{R}^N, \forall k$, then we have*

$$\Delta^{(r+1)} \leq 3 \max \left\{ L, 2 \log^2(2NK) \left( L_{\max} + \frac{L^2}{L_{\min}} \right) \right\} \frac{R_0^2}{r+1}. \tag{17}$$

We comment on the bounds derived in the above theorem. The bound for BCPG with uniform "conservative" stepsize $1/L$ has the same order as the GD method, except for the $\log^2(2NK)$ factor (cf. (7)). In [4, Corollary 3.2], it is shown that the BCGD with the same "conservative" stepsize achieves a sublinear rate with a constant of $4L(1+K)R_0^2$, which is about $K/(3\log^2(2NK))$ times worse than our bound. Further, our bound has the same dependency on $L$ (i.e., $12L$ v.s. $L/2$) as the one derived in [18] for BCPG with a "conservative" stepsize to solve an $\ell_1$ penalized quadratic problem with special data matrix, but our bound holds true for a much larger class of problems (i.e., all quadratic nonsmooth problem in the form of (8)). However, in practice such conservative stepsize is slow (compared with BCPG with $P_k = L_k$, for all $k$) hence is rarely used.

The rest of the bounds derived in Theorem 2.1 is again at least $K/\log^2(2NK)$ times better than existing bounds of cyclic BCPG. For example, when the problem is smooth and unconstrained, the ratio between our bound (17) and the bound (6) is given by

$$\frac{6R_0^2\log^2(2NK)(L^2/L_{\min}+L_{\max})}{CL_{\max}(1+KL^2/L_{\min}^2)R_0^2} \leq \frac{6\log^2(2NK)(1+L^2/(L_{\min}L_{\max}))}{C(1+KL^2/L_{\min}^2)} = \mathcal{O}(\log^2(2NK)/K)$$

(18)

where in the last inequality we have used the fact that $L_{\max}/L_{\min} \geq 1$.

For unconstrained smooth problems, let us compare the bound derived in the second part of Theorem 2.1 (stepsize $P_k = L_k, \forall k$) with that of the GD (7). If $L = KL_k$ for all $k$ (problem badly conditioned), our bound is about $K\log^2(2NK)$ times worse than that of the GD. This indicates a counter-intuitive phenomenon: by choosing conservative stepsize $P_k = L, \forall k$ the iteration complexity of BCGD is $K$ times better compared with choosing a more aggressive stepsize $P_k = L_k, \forall k$. It also indicates that the factor $L/L_{\min}$ may hide an additional factor of $K$.

## 3 Iteration Complexity for General Convex Problems

In this section, we consider improved iteration complexity bounds of BCD for general unconstrained smooth convex problems. We prove a general iteration complexity result, which includes a result of Beck et al. [4] as a special case. Our analysis for the general case also applies to smooth quadratic problems, but is very different from the analysis in previous sections for quadratic problems. For simplicity, we only consider the case $N = 1$ (scalar blocks); the generalization to the case $N > 1$ is left as future work.

Let us assume that the smooth objective $g$ has second order derivatives $H_{ij}(x) := \frac{\partial^2 g}{\partial x_i \partial x_j}(x)$. When each block is just a coordinate, we assume $|H_{ij}(x)| \leq L_{ij}, \forall i,j$. Then $L_i = L_{ii}$ and $L_{ij} \leq \sqrt{L_i}\sqrt{L_j}$. For unconstrained smooth convex problems with scalar block variables, the BCPG iteration reduces to the following coordinate gradient descent (CGD) iteration:

$$x^{(r)} = w_1^{(r)} \xrightarrow{d_1} w_2^{(r)} \xrightarrow{d_2} w_3^{(r)} \longrightarrow \dots \xrightarrow{d_K} w_{K+1}^{(r)} = x^{(r+1)},$$

(19)

where $d_k = \nabla_k g(w_k^{(r)})$ and $w_k^{(r)} \xrightarrow{d_k} w_{k+1}^{(r)}$ means that $w_{k+1}^{(r)}$ is a linear combination of $w_k^{(r)}$ and $d_k e_k$ ($e_k$ is the $k$-th block unit vector).

In the following theorem, we provide an iteration complexity bound for the general convex problem. The proof framework follows the standard three-step approach that combines sufficient descent and cost-to-go estimate; nevertheless, the analysis of the sufficient descent is very different from the methods used in the previous sections. The intuition is that CGD can be viewed as an inexact gradient descent method, thus the amount of descent can be bounded in terms of the norm of the full gradient. It would be difficult to further tighten this bound if the goal is to obtain a sufficient descent based on the norm of the full gradient. Having established the sufficient descent in terms of the full gradient $\nabla g(x^{(r)})$, we can easily prove the iteration complexity result, following the standard analysis of GD (see, e.g. [11, Theorem 2.1.13]).

**Theorem 3.1.** *For CGD with $P_k \geq L_{\max}, \forall k$, we have*

$$g(x^{(r)}) - g(x^*) \leq 2\left(P_{\max} + \frac{\min\{KL^2, (\sum_k L_k)^2\}}{P_{\min}}\right)\frac{R_0^2}{r}, \ \forall\, r \geq 1.$$

(20)

**Proof.** Since $w_{k+1}^r$ and $w_k^r$ only differ by the $k$-th block, and $\nabla_k g$ is Lipschitz continuous with Lipschitz constant $L_k$, we have [2]

$$
\begin{aligned}
g(w_{k+1}^r) \leq &g(w_k^r) + \langle \nabla_k g(w_k^r), w_{k+1}^r - w_k^r \rangle + \frac{L_k}{2}\|w_{k+1}^r - w_k^r\|^2 \\
= &g(w_k^r) - \frac{2P_k - L_k}{2P_k^2}\|\nabla_k g(w_k^r)\|^2 \\
\leq &g(w_k^r) - \frac{1}{2P_k}\|\nabla_k g(w_k^r)\|^2,
\end{aligned}
\tag{21}
$$

where the last inequality is due to $P_k \geq L_k$.

The amount of decrease can be estimated as

$$
g(x^r) - g(x^{r+1}) = \sum_{k=1}^r [g(w_k^r) - g(w_{k+1}^r)] \geq \sum_{k=1}^r \frac{1}{2P_k}\|\nabla_k g(w_k^r)\|^2.
\tag{22}
$$

Since

$$
w_k^r = x^r - \left[\tfrac{1}{P_1}d_1, \ldots, \tfrac{1}{P_{k-1}}d_{k-1}, 0, \ldots, 0\right]^T,
$$

by the mean-value theorem, there must exist $\xi_k$ such that

$$
\begin{aligned}
\nabla_k g(x^r) - \nabla_k g(w_k^r) &= \nabla(\nabla_k g)(\xi_k) \cdot (x^r - w_k^r) \\
&= \left[\frac{\partial^2 g}{\partial x_k \partial x_1}(\xi_k), \ldots, \frac{\partial^2 g}{\partial x_k \partial x_{k-1}}(\xi_k), 0, \ldots, 0\right]\left[\tfrac{1}{P_1}d_1, \ldots, \tfrac{1}{P_{k-1}}d_{k-1}, 0, \ldots, 0\right]^T \\
&= \left[\frac{1}{\sqrt{P_1}}H_{k1}(\xi_k), \ldots, \frac{1}{\sqrt{P_{k-1}}}H_{k,k-1}(\xi_k), 0, \ldots, 0\right]\left[\tfrac{1}{\sqrt{P_1}}d_1, \ldots, \tfrac{1}{\sqrt{P_K}}d_K\right]^T,
\end{aligned}
\tag{23}
$$

where $H_{ij}(x) = \frac{\partial^2 g}{\partial x_i \partial x_j}(x)$ is the second order derivative of $g$. Then

$$
\begin{aligned}
\nabla_k g(x^r) &= \nabla_k g(x^r) - \nabla_k g(w_k^r) + \nabla_k g(w_k^r) \\
&= \left[\frac{1}{\sqrt{P_1}}H_{k1}(\xi_k), \ldots, \frac{1}{\sqrt{P_{k-1}}}H_{k,k-1}(\xi_k), 0, \ldots, 0\right]\left[\tfrac{1}{\sqrt{P_1}}d_1, \ldots, \tfrac{1}{\sqrt{P_K}}d_K\right]^T + d_k \\
&= \left[\frac{1}{\sqrt{P_1}}H_{k1}(\xi_k), \ldots, \frac{1}{\sqrt{P_{k-1}}}H_{k,k-1}(\xi_k), \sqrt{P_k}, 0, \ldots, 0\right]\left[\tfrac{1}{\sqrt{P_1}}d_1, \ldots, \tfrac{1}{\sqrt{P_K}}d_K\right]^T \\
&= v_k^T d,
\end{aligned}
\tag{24}
$$

where we have defined

$$
\begin{aligned}
d &:= \left[\tfrac{1}{\sqrt{P_1}}d_1, \ldots, \tfrac{1}{\sqrt{P_K}}d_K\right]^T, \\
v_k &:= \left[\frac{1}{\sqrt{P_1}}H_{k1}(\xi_k), \ldots, \frac{1}{\sqrt{P_{k-1}}}H_{k,k-1}(\xi_k), \sqrt{P_k}, \ldots, 0\right].
\end{aligned}
\tag{25}
$$

Let

$$
V := \begin{bmatrix} v_1^T \\ \ldots \\ v_K^T \end{bmatrix} = \begin{bmatrix}
\sqrt{P_1} & 0 & 0 & \ldots & 0 & 0 \\
\frac{1}{\sqrt{P_1}}H_{21}(\xi_2) & \sqrt{P_2} & 0 & \ldots & 0 & 0 \\
\frac{1}{\sqrt{P_1}}H_{31}(\xi_3) & \frac{1}{\sqrt{P_2}}H_{32}(\xi_3) & \sqrt{P_3} & \ldots & 0 & 0 \\
\frac{1}{\sqrt{P_1}}H_{41}(\xi_4) & \frac{1}{\sqrt{P_2}}H_{42}(\xi_4) & \frac{1}{\sqrt{P_3}}H_{43}(\xi_4) & \ddots & 0 & 0 \\
\vdots & \vdots & \vdots & \vdots & \ddots & \vdots \\
\frac{1}{\sqrt{P_1}}H_{K1}(\xi_K) & \frac{1}{\sqrt{P_2}}H_{K2}(\xi_K) & \frac{1}{\sqrt{P_3}}H_{K3}(\xi_K) & \ldots & \frac{1}{\sqrt{P_{K-1}}}H_{K,K-1}(\xi_K) & \sqrt{P_K}
\end{bmatrix}
\tag{26}
$$

Therefore, we have

$$\|\nabla g(x^r)\|^2 = \sum_k \|\nabla_k g(x^r)\|^2 \stackrel{(24)}{=} \sum_k \|v_k^T d\|^2 = \|Vd\|^2 \leq \|V\|^2 \|d\|^2 = \|V\|^2 \sum_k \frac{1}{P_k} \|\nabla_k g(w_k^r)\|^2.$$

Combining with (22), we get

$$g(x^r) - g(x^{r+1}) \geq \sum_k \frac{1}{2P_k} \|\nabla_k g(w_k^r)\|^2 \geq \frac{1}{2\|V\|^2} \|\nabla g(x^r)\|^2. \tag{27}$$

Let $D \triangleq \mathrm{Diag}(P_1, \ldots, P_K)$ and let $H(\xi)$ be defined as

$$H(\xi) := \begin{bmatrix} 0 & 0 & 0 & \ldots & 0 & 0 \\ H_{21}(\xi_2) & 0 & 0 & \ldots & 0 & 0 \\ H_{31}(\xi_3) & H_{32}(\xi_3) & 0 & \ldots & 0 & 0 \\ \vdots & \vdots & \vdots & \vdots & \ddots & \vdots \\ H_{K1}(\xi_K) & H_{K2}(\xi_K) & H_{K3}(\xi_K) & \ldots & H_{K,K-1}(\xi_K) & 0 \end{bmatrix}. \tag{28}$$

Then $V = D^{1/2} + H(\xi)D^{-1/2}$, which implies

$$\|V\|^2 = \|D^{1/2} + H(\xi)D^{-1/2}\|^2 \leq 2(\|D^{1/2}\|^2 + \|H(\xi)D^{-1/2}\|^2) \leq 2\left(P_{\max} + \frac{\|H(\xi)\|^2}{P_{\min}}\right).$$

Plugging into (27), we obtain

$$g(x^{(r)}) - g(x^{(r+1)}) \geq \frac{1}{2} \frac{1}{P_{\max} + \frac{\|H(\xi)\|^2}{P_{\min}}} \|\nabla g(x^{(r)})\|^2. \tag{29}$$

From the fact that $H_{kj}(\xi_k)$ is a scalar bounded above by $|H_{kj}(\xi_k)| \leq L_{kj} \leq \sqrt{L_k L_j}$, thus

$$\|H\|^2 \leq \|H\|_F^2 = \sum_{k<j} |H_{kj}(\xi_k)|^2 \leq \sum_{k<j} L_k L_j \leq \left(\sum_k L_k\right)^2. \tag{30}$$

We provide the second bound of $\|H\|$ below. Let $H_k$ denote the $k$-th row of $H$, then $\|H_k\| \leq L$. Therefore, we have

$$\|H\|^2 \leq \|H\|_F^2 = \sum_k \|H_k\|^2 \leq \sum_k L^2 = KL^2.$$

Combining this bound and (30), we obtain that $\|H\|^2 \leq \min\{KL^2, (\sum_k L_k)^2\} \triangleq \beta^2$.

Denote $\omega = \frac{1}{2} \frac{1}{P_{\max} + \frac{\beta^2}{P_{\min}}}$, then (29) becomes

$$g(x^{(r)}) - g(x^{(r+1)}) \geq \omega \|\nabla g(x^{(r)})\|^2, \ \forall r. \tag{31}$$

This relation also implies $g(x^{(r)}) \leq g(x^{(0)})$, thus by the definition of $R_0$ in (3) we have $\|x^{(r)} - x^*\| \leq R_0$. By the convexity of $g$ and the Cauchy-Schwartz inequality, we have

$$g(x^{(r)}) - g(x^*) \leq \langle \nabla g(x^{(r)}), x^{(r)} - x^* \rangle \leq \|\nabla g(x^{(r)})\| R_0.$$

Combining with (31), we obtain

$$g(x^{(r)}) - g(x^{(r+1)}) \geq \frac{\omega}{R_0^2} (g(x^{(r)}) - g(x^*))^2.$$

Let $\Delta^{(r)} = g(x^{(r)}) - g(x^*)$, we obtain

$$\Delta^{(r)} - \Delta^{(r+1)} \geq \frac{\omega}{R_0^2} \Delta^{(r)}.$$

Then we have

$$\frac{1}{\Delta^{(r+1)}} \geq \frac{1}{\Delta^{(r)}} + \frac{\omega}{R_0^2} \frac{\Delta^{(r)}}{\Delta^{(r+1)}} \geq \frac{1}{\Delta^{(r)}} + \frac{\omega}{R_0^2}.$$

Summarizing the inequalities, we get

$$\frac{1}{\Delta^{(r+1)}} \geq \frac{1}{\Delta^{(0)}} + \frac{\omega}{R_0^2}(r+1) \geq \frac{\omega}{R_0^2}(r+1),$$

which leads to

$$\Delta^{(r+1)} = g(x^{(r+1)}) - g(x^*) \leq \frac{1}{\omega}\frac{R_0^2}{r+1} = 2(P_{\max} + \frac{\beta^2}{P_{\min}})\frac{R_0^2}{r+1},$$

where $\beta^2 = \min\{KL^2, (\sum_k L_k)^2\}$. This completes the proof. **Q.E.D.**

Let us compare this bound with the bound derived in [4, Theorem 3.1] (replacing the denominator $r + 8/K$ by $r$), which is

$$g(x^r) - g(x^*) \leq 4\left(P_{\max} + \frac{P_{\max}}{P_{\min}}\frac{KL^2}{P_{\min}}\right)\frac{R^2}{r}. \tag{32}$$

In our new bound, besides reducing the coefficient from 4 to 2 and removing the factor $\frac{P_{\max}}{P_{\min}}$, we improve $KL^2$ to $\min\{KL^2, (\sum_k L_k)^2\}$. Neither of the two bounds $KL^2$ and $(\sum_k L_k)^2$ implies the other: when $L = L_k, \forall k$ the new bound $(\sum_k L_k)^2$ is $K$ times larger; when $L = KL_k, \forall k$ or $L = L_1 > L_2 = \cdots = L_K = 0$ the new bound is $K$ times smaller. In fact, when $L = KL_k, \forall k$, our new bound is $K$ times better than the bound in [4] for either $P_k = L_k$ or $P_k = L$. For example, when $P_k = L, \forall k$, the bound in [4] becomes $\mathcal{O}(\frac{KL}{r})$, while our bound is $\mathcal{O}(\frac{L}{r})$, which matches GD (listed in Table 1 below). Another advantage of the new bound $(\sum_k L_k)^2$ is that it does not increase if we add an artificial block $x_{K+1}$ and perform CGD for function $\tilde{g}(x, x_{K+1}) = g(x)$; in contrast, the existing bound $KL^2$ will increase to $(K+1)L^2$, even though the algorithm does not change at all.

We have demonstrated that our bound can match GD in some cases, but can possibly be $K$ times worse than $GD$. An interesting question is: for general convex problems can we obtain an $\mathcal{O}(\frac{L}{r})$ bound for cyclic BCGD, matching the bound of GD? Removing the $K$-factor in (32) will lead to an $\mathcal{O}(\frac{L}{r})$ bound for conservative stepsize $P_k = L$ no matter how large $L_k$ and $L$ are. We conjecture that an $\mathcal{O}(\frac{L}{r})$ bound for cyclic BCGD cannot be achieved for general convex problems. That being said, we point out that the iteration complexity of cyclic BCGD may depend on other intrinsic parameters of the problem such as $\{L_k\}_k$ and, possibly, third order derivatives of $g$. Thus the question of finding the best iteration complexity bound of the form $\mathcal{O}(h(K)\frac{L}{r})$, where $h(K)$ is a function of $K$, may not be the right question to ask for BCD type algorithms.

## 4   Conclusion

In this paper, we provide new analysis and improved complexity bounds for cyclic BCD-type methods. For convex quadratic problems, we show that the bounds are $\mathcal{O}(\frac{L}{r})$, which is independent of $K$ (except for a mild $\log^2(2K)$ factor) and is about $L_{\max}/L + L/L_{\min}$ times worse than those for GD/PG. By a simple example we show that it is not possible to obtain an iteration complexity $\mathcal{O}(L/(Kr))$ for cyclic BCPG. For illustration, the main results of this paper in several simple settings are summarized in the table below. Note that different ratios of $L$ over $L_k$ can lead to quite different comparison.

**Table 1:** Comparison of Various Iteration Complexity Results

| Lip-constant 1/Stepsize | Diagonal Hessian $L_i = L$ $P_i = L$ | Full Hessian $L_i = \frac{L}{K}$ Large stepsize $P_i = \frac{L}{K}$ | Full Hessian $L_i = \frac{L}{K}$ Small stepsize $P_i = L$ |
|---|---|---|---|
| GD | $L/r$ | N/A | $L/r$ |
| Random BCGD | $L/r$ | $L/(Kr)$ | $L/r$ |
| Cyclic BCGD [4] | $KL/r$ | $K^2L/r$ | $KL/r$ |
| Cyclic CGD, Cor 3.1 | $KL/r$ | $KL/r$ | $L/r$ |
| Cyclic BCGD (QP) | $\log^2(2K)L/r$ | $\log^2(2K)KL/r$ | $\log^2(2K)L/r$ |

## Footnotes

[1]Note that the assumptions made in [4] and [7] are slightly different, but the rates derived in both cases have similar dependency on the problem dimension $K$.

[2] A stronger bound is $g(w_{k+1}^r) \leq g(w_k^r) - \frac{1}{2\hat{P}_k}\|\nabla_k g(w_k^r)\|^2$, where $\hat{P}_k = \frac{P_k^2}{2P_k - L_k} \leq P_k$, but since $P_k \leq 2P_k - L_k \leq 2P_k$, the improvement ratio of using this stronger bound is no more than a factor of 2.

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
