[Supplementary Material]

# Appendix

## 4.1 Proof of Lemma 2.2

**Proof.** First note that $g(x)$ being convex quadratic implies that its second order Taylor expansion is tight

$$g(x^*) - g(x^{(r+1)}) = \langle \nabla g(x^{(r+1)}), x^* - x^{(r+1)} \rangle + \frac{1}{2}\|A(x^{(r+1)} - x^*)\|^2. \tag{33}$$

Using this fact we can estimate $f(x^{(r+1)}) - f(x^*)$ by the following series of inequalities

$f(x^{(r+1)}) - f(x^*)$

$= \left\langle \nabla g(x^{(r+1)}), x^{(r+1)} - x^* \right\rangle + h(x^{(r+1)}) - h(x^*) - \frac{1}{2}\|A(x^{(r+1)} - x^*)\|^2$

$\overset{(i)}{\leq} \sum_{k=1}^{K} \left\langle \nabla_k g(x^{(r+1)}) - \nabla_k g(w_k^{(r+1)}), x_k^{(r+1)} - x_k^* \right\rangle - \sum_{k=1}^{K} P_k \left\langle x_k^{(r+1)} - x_k^{(r)}, x_k^{(r+1)} - x_k^* \right\rangle - \frac{1}{2}\|A(x^{(r+1)} - x^*)\|^2$

$= \sum_{k=1}^{K} \left\langle \left(\sum_{j \geq k} A_j(x_j^{(r+1)} - x_j^{(r)})\right), A_k(x_k^{(r+1)} - x_k^*) \right\rangle - (x^{(r+1)} - x^{(r)})^T (\widetilde{P} \otimes I_N)(x^{(r+1)} - x^*)$

$\quad - \frac{1}{2}\|A(x^{(r+1)} - x^*)\|^2$

$= (x^{(r+1)} - x^{(r)})^T \widetilde{A}^T (D_1 \otimes I_M)\widetilde{A}(x^{(r+1)} - x^*) - (x^{(r+1)} - x^{(r)})^T (\widetilde{P} \otimes I_N)(x^{(r+1)} - x^*)$

$\quad - \frac{1}{2}\|A(x^{(r+1)} - x^*)\|^2$

$\leq (x^{(r+1)} - x^{(r)})^T \left( \widetilde{A}^T (D_1 \otimes I_M)\widetilde{A} - \widetilde{P} \otimes I_N \right) (x^{(r+1)} - x^*) \tag{34}$

where in (i) we have used the optimality condition of the subproblem (9) (i.e., (11)); in the last equality we have defined a lower triangular matrix $D_1$

$$D_1 := \begin{bmatrix} 1 & 0 & 0 & \cdots & 0 & 0 \\ 1 & 1 & 0 & \cdots & 0 & 0 \\ 1 & 1 & 1 & \cdots & 0 & 0 \\ \vdots & \vdots & \vdots & \cdots & \vdots & \vdots \\ 1 & 1 & 1 & \cdots & 1 & 1 \end{bmatrix} \in \mathbb{R}^{K \times K}. \tag{35}$$

Notice that the following is true

$$\widetilde{A}^T (D_1 \otimes I_M)\widetilde{A} = \left( A^T A - \widetilde{A}^T \widetilde{A} \right) \odot D_2 + \widetilde{A}^T \widetilde{A},$$

where "$\odot$" denotes the Hadamard product; $D_2$ is a lower triangular matrix similarly as defined in (35), but of dimension $KN \times KN$. Combining this identity and (34), we have

$\Delta^{(r+1)} \leq \left( (x^{(r+1)} - x^{(r)})(\widetilde{P}^{1/2} \otimes I_N) \right)^T \left( (\widetilde{P}^{-1/2} \otimes I_N)(A^T A - \widetilde{A}^T \widetilde{A}) \odot D_2 \right.$

$\left. \quad + (\widetilde{P}^{-1/2} \otimes I_N)\widetilde{A}^T \widetilde{A} - \widetilde{P}^{1/2} \otimes I_N \right)(x^{(r+1)} - x^*)$

$\overset{(i)}{\leq} \left\| (x^{(r+1)} - x^{(r)})(\widetilde{P}^{1/2} \otimes I_N) \right\| \left\| (\widetilde{P}^{-1/2} \otimes I_N)(A^T A - \widetilde{A}^T \widetilde{A}) \odot D_2 \right\| \|x^{(r+1)} - x^*\|$

$\overset{(ii)}{\leq} \left\| (x^{(r+1)} - x^{(r)})(\widetilde{P}^{1/2} \otimes I_N) \right\| \left\| (\widetilde{P}^{-1/2} \otimes I_N)(A^T A - \widetilde{A}^T \widetilde{A}) \right\| \left(1 + \frac{1}{\pi} + \frac{\log(NK)}{\pi}\right) R_0$

$\overset{(iii)}{\leq} R_0 \left\| (x^{(r+1)} - x^{(r)})(\widetilde{P}^{1/2} \otimes I_N) \right\| \left\| (\widetilde{P}^{-1/2} \otimes I_N)(A^T A - \widetilde{A}^T \widetilde{A}) \right\| \log(2NK)$

$\leq R_0 \log(2NK) \left(L/\sqrt{P_{\min}} + \sqrt{P_{\max}}\right) \left\| (x^{(r+1)} - x^{(r)})(\widetilde{P}^{1/2} \otimes I_N) \right\| \tag{36}$

where (i) uses the Cauchy-Schwartz inequality and the fact that $\widetilde{P} \otimes I_N \succeq \widetilde{A}^T \widetilde{A}$; (iii) is true for all $KN \geq 3$. Inequality (ii) is true due to a result on the spectral norm of the triangular truncation operator; see [1, Theorem 1]. In particular, Define

$$Y_{KN}(D_2) = \max \left\{ \frac{\|Z \odot D_2\|}{\|Z\|} : Z \in \mathbb{R}^{KN \times KN}, Z \neq 0 \right\}.$$

Then we have the following estimate

$$\left| \frac{Y_{KN}(D_2)}{\log(KN)} - \frac{1}{\pi} \right| \leq \left( 1 + \frac{1}{\pi} \right) \frac{1}{\log(KN)}.$$

The proof is completed. **Q.E.D.**

## 4.2 Proof of Theorem 2.1

**Proof.** For notational simplicity, let us define

$$C := R_0 \log(2NK) \left( L/\sqrt{P_{\min}} + \sqrt{P_{\max}} \right).$$

Taking a square of the cost-to-go estimate (14) and the sufficient descent estimate (12), we obtain

$$(\Delta^{(r+1)})^2 \leq C^2 \left\| (x^{(r+1)} - x^{(r)})(\widetilde{P}^{1/2} \otimes I_N) \right\|^2$$

$$= C^2 \sum_{k=1}^{K} P_k \|x_k^{(r+1)} - x_k^{(r)}\|^2$$

$$\leq 2C^2 (\Delta^{(r)} - \Delta^{(r+1)}).$$

Utilizing a result from [2, Lemma 3.5], the above inequality implies that

$$\Delta^{(r+1)} \leq 3 \max \left\{ \Delta^0, 2C^2 \right\} \frac{1}{r+1}$$

$$\leq 3 \max \left\{ \Delta^0, 2 \log^2(2NK) \left( P_{\max} + \frac{L^2}{P_{\min}} \right) R_0^2 \right\} \frac{1}{r+1} \tag{37}$$

When $P_k = L$ for all $k$, the bound reduces to

$$\Delta^{(r+1)} \leq 3 \max \left\{ \Delta^0, 4 \log^2(2NK) R_0^2 L \right\} \frac{1}{r+1} \tag{38}$$

When the problem is smooth and unconstrained, we have

$$\Delta^{(0)} \leq \langle \nabla f(x^{(0)}), x^{(0)} - x^* \rangle = \langle \nabla f(x^{(0)}) - f(x^*), x^{(0)} - x^* \rangle \leq L \|x^{(0)} - x^*\|^2.$$

This completes the proof. **Q.E.D.**