[Reviews · NeurIPS 2015]

Submitted by Assigned_Reviewer_1

Summary:

The authors present an improved iteration complexity of Block Coordinate Descent (BCD) typed algorithm for composite minimization problems. When the smooth part is quadratic, the authors prove an improved iteration complexity without dependency to K (number of blocks or coordinates), and show that the bound is tight in terms of K. For general composite functions, the authors also derived a new iteration bound. This new bound is better than [3] in some cases.

Comments:

Pros:

- The theoretical results and the proof strategies are presented clearly.

- The iteration complexity for the quadratic case is pretty tight and has a significant improvement over

the bound in [3].

Cons:

- The new bound for general convex problems is less interesting since it is unlikely to have a problem with L=KL. Probably the authors can compute this bound for some datasets to show the improvement. So the main improvement is only for the quadratic case.

Summary: A well written paper. The authors significantly improve the iteration complexity for cyclic coordinate descent when solving the quadratic problem with convex regularization.

Submitted by Assigned_Reviewer_2

The paper improves the existing analysis of cyclic block coordinate

methods, and establishes convergance rates that matches those of

randomized techniques for convex problems. The paper starts by

analyzing optimization problems involving a convex loss, before

deriving more general results for smooth unconstrained problems.

The paper is very clear and easy to read, despite its theoretical

nature. It provides a result that closes a gap in the

literature, given the fact that most efforts have been recently

devoted to randomized algorithms.

Overall, I like the paper. My only regret is the the lack of

a discussion including a comparison with existing

results for randomized proximal coordinate descent techniques. This

would settle the question of the importance of randomization, which

is often chosen for the simplicity of its convergence proof.

I believe that the paper makes an important contribution by

settling this question. A discussion about possible extensions of the

analysis to strongly-convex problems would be nice as well.

Misc:

- please define R_0 in (6) to make the paper self-content.

- page 3,

blkdlg(L_k) should be blkdlg(P_k)

Summary: The paper provides new insight why cyclic coordinate descent

performs very well for some classes of problems. It provides a

theoretical explanation for a known empirical phenomenon.

Submitted by Assigned_Reviewer_3

Report for "Improved Iteration Complexity Bounds of Cyclic Block Coordinate Descent for Convex Problems".

## Summary

The authors consider methods of block coordinate types with cyclic block selection rules. The focus is on a composite model involving a non-separable smooth term and a separable non-smooth term. The main methods of interest are block proximal gradient and cyclic minimization. The authors propose a refined analysis of these methods for problems for which the smooth term is quadratic. This leads to new convergence rate estimates for this class of problems. Finally an alternative analysis is presented for the general case with the restriction that blocks should have size one (coordinate proximal gradient).

## Main comments

The analysis of sections 2 is original, relevant and quite impressive. In some settings, the complexity estimates match those of the usual proximal gradient method which was not known to be possible for such a general class of problems. The analysis removes a multiplicative dependance in the number of blocks compared to previous work. This constitutes a very strong theoretical contribution. Section 3 is also very interesting, but I have one concern with the proof of Theorem 3.1 (see next paragraph). The results of section 4 are a bit less spectacular but still of great interest. Overall, the proof arguments look reasonable and are quite original. As the questions adressed in this paper are strongly related to both machine learning and optimization, I warmly recommend this work for publication after taking into account the following remark:

In proof of Theorem 3.1 at the

bottom of page 15, I think that the last step (i) is wrong. It is mentioned that (i) can be derived from (49), but I am not sure it is true since Lmin is not the minimal eigenvalue of the matrix of interest in (49), but the minimum of the maximum eigenvalues of its diagonal blocks. I think it is possible to fix this by introducing a factor K, using

| | sum b _ i | | ^ 2 leq K sum | | b _ i | | ^2

It could also be possible to fix it by using the idea that exact minimization makes more progress than a proximal gradient step, and using arguments from Lemma 2.1, but I am not completely sure about it.

In any case, if the authors agree with this remark, important changes have to be made in the proofs and/or in the main text.

## Additional comments and suggestions

- An important variant of cyclic block decomposition methods is the random permutation approach where the order of blocks changes at each itteration. In some applications, this additional randomization gives a practical advantage. Most of results usually given for cyclic block decomposition methods still hold deterministically when the order of the blocks changes arbitrarily between iterations. I suspect that this is also the case for the proposed analysis which, if true, would be worth mentioning.

- In section 4, the assumption that N=1 is not only for notational simplicity. The analysis falls appart when N > 1. Indeed, the analysis uses the mean valued-theorem in equation (78) which does not hold for vector valued functions. Therefore the analysis cannot be straightforwardly extended to blocks of size more than one without further arguments.

- The quantity defined in (3) might very well be infinite, unless compactness of sublevel sets is assumed as in [3] for example. This could be precised.

- Notation in (2) constitutes a slight abuse as the order of the arguments of g is not the same as in (1). Without needing to change notations, it could be a good idea to give precisions.

- I do not think there is a result as (6) in [3] which is concerned mostly about BCGD and not BCD (only for the two block case). Maybe also give a precise pointer in reference [6].

- In (8), all the blocks are assumed to have the same size. I guess the analysis can be extended to cases for which blocks have different sizes. Maybe emphasize that this is not an assumption made for restriction, but for simplicity.

- In (7), the constant is independant of problem dimension for fixed ||x^0 - x^*|| and L. From a practical perspective, these values usually depend on the dimension of the problem.

- (11) is true, although not very common. Maybe add a sentence explaining that both (9) and (11) are equivalent to the same subdifferential inclusion.

- Kroneker - > Kronecker. Maybe precise the notation. I suspect that in the notations of the paper, the order of matrices involved in Kronecker products is reversed compared to other sources (e.g. wikipedia).

- Line 661 and 688 in the appendix refer to optimality condition of w_{k+1}^{r+1}. I guess it should be optimality of x^(r)_{k}. Furthermore, this optimality condition could be written explicitely as (11) is for example.

- Just before Thorem 4.1, it is mentioned that the results can be established "in the same way as the standard analysis of GD". It would be nicer to have the precise statement, and at least a reference (e.g. [Theorem 2.1.13 10]).

- Quantification over r could be more precise in Theorem 4.1 and Corollary 4.1, in particylar, the estimates presented are not well defined for r = 0.

- In the discussion of section 4, I think that another relevant feature of the newly proposed bound is that it does not increase with the artificial addition of new variables on which the problem does not depend (with Li = 0). This is not the case for the quantity KL which is problematic since such an artificial introduction does not modify the iterates an we therefore anticipate that it should not modify the complexity estimates.

- In the abstract, it is worth precising that the iteration counter is the number of times all blocks have been updated. It would be OK to have a dependance in K if the counter consisted in the number of blocks updated as in Random block methods.

- Page 5, line 228, I guess "same dependency on K" should be "same dependency on L".

- Remark 3.1 "those derive here" - > "those derived here".

- Toplitz - > Toeplitz.

- Although transparent, the notations Pmax and Pmin are never defined.

- In Lemma 4.1, the logical quantification over xi should be more precise. I think that the correct quantifier is "there exists"

- In (36), I guess there are missing indices for A(:,i). This is matlab notation, maybe precise what it means.

- In the

proof of Lemma 2.1, there are missing indices k in $h_k(x^r)$ on line 504

- In (45), there is a use of (11) on the fifth line which could be precised.

- Line 637 "the fact" - > "and the fact"

- Line 702 "Combining the above to inequalities". There are one inequality and one equality. "to" - > "two".

- Section 13 should be "Proof of Corrolary 4.1"?
Summary: The analysis of sections 2 is original, relevant and quite impressive. I warmly recommend this work for publication after taking into account an important remark.

Submitted by Assigned_Reviewer_4

The motivation of the paper is, indeed, interesting. They try to get rid of the dimension dependence in the convergence analysis. However, maybe it is a bit hidden in the global lip. constant L. It can be shown that there exists and example, when L_min = L_max and moreover L = K * L_max. (see Lemma 2.1. in http://epubs.siam.org/doi/abs/10.1137/140964795 ) Moreover, in Corollary 3.3. of that paper they showed a complexity result of O(sqrt{K} L/epsilon), however, for a special problems it can be improved to O(L/epsilon).

The proofs probably contains many problems, let me just mention proof of Lemma 2.1 1. the first equality (line 499) is OK and clear 2. the first inequality is not valid (line 502). one more summation for h_k is needed, otherwise it will not cancel out in the next step 3. line 509: change "equality" to "inequality" Otherwise the statement of Lemma 2.1 should be valid, just the proofs and papers are not written carefully. Also note that h_k: R^N -> R and x^{r} is actually from \R^{K N}, hence the expression h_k (x^{r}) has no meaning and should be replaced by h_k (x^{r}_k)

Another issue is e.g. on line 158. One should write \tilde P Kroneker product with I_N and not as stated.

Line 151, the matrix \tilde A has dimension MK x NK

Summary: The paper is providing improved complexity analysis or block coordinate descent method. Authors proposed a modified version of convergence theory which combines 3 steps: estimate descent of the objective; estimate how much we can improve; and combine those two estimates. If the proofs are correct, then this result would be very nice.

Author Feedback
Author rebuttal: Response to Review 1.
Thanks. The reviewer said "it is unlikely to have a problem with L=KL_i." In fact, L/L_i will be large if the Hessian has large off_diagonal entries and it seems quite possible to have large L/L_i (though may not be K). In the revised paper, we have added another example that our new bound (sum_i L_i)^2 is K times better than the old bound KL^2: unbalanced case L = L_1 > L_2=...=L_K = 0 (sometimes this case can be avoided by normalization, but this example still shows how bad the existing bound KL^2 can be). In the journal version of the paper we will consider your suggestion and test a few real data sets to compare the two bounds KL^2 and (sum_i L_i)^2.

Response to Review 2.
Thanks. In the revised paper we have included some discussion about the known results for randomized BCPG algorithms. We have also incorporated a few other suggestions you mentioned. In particular:
1) R0 in (6) has been defined in (3) in the original paper.
2) We have changed L_k to P_k
Response to Review 3.
We thank the reviewer for the detailed comments. Please see below for our response.
1) The comment on the last equation on page 15 was right. We now used the following bound
||A(x^{t}-x^{t+1})||\le K ||\tilde{A}( x^{t}-x^{t+1})|| instead what we have used below. We have also revised a few places about the exact BCD.
2) The reviewer is correct about the permutation scheme. Our analysis indicates that the random permutation (sample without replacement) convergence deterministically, with the same rate. We have mentioned in the revised paper about the complexity of random permutation schemes.
3) About the question whether the results in section 4 can be extended to N>1, we have removed "for notational simplicity" and added "the extension is left as future work". We have a way to resolve the issue the reviewer mentioned, but due to the time limit we have not fully checked the correctness of the full proof for the block case.
4) About the little abuse of notation in (2), we have added a sentence to clarify that the change of arguments does not affect the function value.
5) (11) is a standard optimality condition in constrained convex optimization, though this condition does not involve subdifferential. We have added a reference before (11) to clarify this point.
6) We have changed the notation regarding to the Kronecker product to the conventional ones.
7) We have added the proof of Thorem 4.1 and the reference as the reviewer suggested.
8) In Theorem 4.1 and Corollary 4.1, we add a condition "r>=1".
9) We have added a sentence to incorporate your comment that adding an artificial block will increase the old bound KL^2.
10) For other typos the reviewer pointed out, we have corrected them in the revised paper.

Response to reviewer 4.
1) The paper by Schmidt and Friedlander does not consider cyclic block coordinate descent method. They consider a Gauss Southwell method in which at each iteration only a single block is updated (in a greedy manner). Also they only consider smooth and strongly convex problem, but in this work we do not impose the strong convexity nor the smoothness assumption on the problem. We have briefly commented this work in the introduction.

Response to reviewer 5.
1) Let us explain why the first inequality in line (502) is correct.
Note that f(x^{r+1}) has K nonsmooth terms h_k(x^{k+1}). In (502), these nonsmooth terms have been distributed into the parenthesis. Indeed, when we collect terms, we can see that (502) has precisely h_k(x^{k+1}), k=1,...K. We do not need any further summation, as suggested by the reviewer.
2) We have replaced all references to h_k(x) to h_k(x_k)
3) We have replaced MK x MK by MK x NK in the definition of \tilde{A}
4) We have rearranged the order of the Kronecker product, as suggested by the reviewer.
Response to reviewer 6.
1) We believe that there is a misunderstanding here. The reviewer mentioned that our result is limited because we only consider "strongly convex quadratic functions". This is definitely not the case. In section 2 and 3, we consider convex problems in which the smooth part is quadratic. Note that in our problem formulation we do have nonsmooth component in the objective and the constraints set. Also we have not mentioned anywhere in the paper about the strong convexity of the problem. Moreover, in section 4, we consider non-quadratic convex problems as well.
2) Yes we agree that the new constant can be as bad as K. However even in such worst case our bounds are still K times better than most of the existing results (cf. Beck et al 13) and (Hong et al 13), in which there is an additional and explicit dependency on K. Also see the remarks after Theorem 2.1. Also when the conservative stepsize is used, our bounds do not depend on (L-Lmin/Lmin), and they match exactly that of the GD.